# BridgePure: Limited Protection Leakage Can Break Black-Box Data Protection

**Yihan Wang**[*,†]
University of Waterloo
yihan.wang@uwaterloo.ca

**Yiwei Lu**[*]
University of Ottawa
yiwei.lu@uottawa.ca

**Xiao-Shan Gao**[‡]
AMSS, Chinese Academy of Sciences
University of Chinese Academy of Sciences
xgao@mmrc.iss.ac.cn

**Gautam Kamath**[‡]
University of Waterloo
Vector Institute
g@csail.mit.edu

**Yaoliang Yu**[‡]
University of Waterloo
Vector Institute
yaoliang.yu@uwaterloo.ca

## Abstract

Availability attacks, or unlearnable examples, are defensive techniques that allow data owners to modify their datasets in ways that prevent unauthorized machine learning models from learning effectively while maintaining the data's intended functionality. It has led to the release of popular black-box tools (e.g., APIs) for users to upload personal data and receive protected counterparts. In this work, we show that such black-box protections can be *substantially compromised* if a small set of unprotected in-distribution data is available. Specifically, we propose a novel threat model of protection leakage, where an adversary can (1) easily acquire (unprotected, protected) pairs by querying the black-box protections with a small unprotected dataset; and (2) train a diffusion bridge model to build a mapping between unprotected and protected data. This mapping, termed *BridgePure*, can effectively remove the protection from any previously unseen data within the same distribution. *BridgePure* demonstrates superior purification performance on classification and style mimicry tasks, exposing critical vulnerabilities in black-box data protection. We suggest that practitioners implement multi-level countermeasures to mitigate such risks. The code is available at https://github.com/EhanW/bridge-pure.

## 1 Introduction

The widespread adoption of machine learning (ML) models has raised significant concerns about data privacy, copyright, and unauthorized use of personal information. Specifically, machine learning developers usually rely on crawling web data to create their training sets, which can result in data being trained on without the owners' consent. This has significant potential for misuse. For example, trained models may be used in sensitive applications such as facial recognition [20], resulting in individual re-identification or serious privacy breaches. Another example is training on copyrighted images created by artists. The downstream models could be used for style mimicry and potentially

---

[*]Equal contribution. [†]Correspondence author. [‡]Listed in alphabetical order.

result in direct copyright infringement in cases where a generative model exactly replicates the same art style as the training data.

Such unauthorized data usage has served as an impetus for broad pushback against the use of ML models. One particular demographic, artists, has been searching for solutions that prevent non-consensual use of their artwork for training ML models. Their desires are somewhat at odds with each other: they would like their artwork to have low value in training an ML model, while simultaneously ensuring that the artwork is of high fidelity to preserve the quality of their original work. This has given rise to a style of availability attack known as "unlearnable examples" [13, 14, 24, 58], wherein imperceptible changes are made to training data points, which nonetheless render them low value for use in ML model training. It has even led to the release of popular tools that serve this or a similar purpose (e.g., Glaze [56], Nightshade [57], and Mist [32]). These offer public APIs (denoted as $\mathcal{P}$) that allow a data owner to input their dataset $\mathcal{D}$ and receive a protected version $\mathcal{D}' = \mathcal{P}(\mathcal{D})$.

We demonstrate that such black-box protection may be susceptible to an attack wherein an adversary can potentially render the protection ineffective. Specifically, given access to a small set $\mathcal{D}_a$ of unprotected in-distribution data (e.g., data collected before protection is deployed; photos taken by others at a party; pictures of art taken at a gallery) and a public protection API $\mathcal{P}$, an adversary can easily acquire $(\mathcal{D}_a, \mathcal{P}(\mathcal{D}_a))$ pairs by querying the black-box service. We call such a risk *protection leakage*. In this paper, we aim to answer an intriguing question:

> *How can protection leakage sabotage data protection? And to what extent?*

Indeed, with a *small* number of pairs, we show that an adversary can easily train a diffusion denoising bridge model (DDBM, [76]) that learns an inverse mapping $\mathcal{P}^{-1}$ such that $\mathcal{P}^{-1}(\mathcal{P}(\mathbf{x})) \approx \mathbf{x}$ for $\mathbf{x} \in \mathcal{D}_a$. Moreover, the learned bridge model generalizes to unseen data from the same distribution and can purify a large amount of protected data, $\mathcal{D}'$. We call this approach **BridgePure**. We show that, with the reasonable assumption of access to a small amount of unprotected in-distribution data, BridgePure gives far better results than prior work [9, 25, 42, 72], without requiring pre-training or fine-tuning a large diffusion model with a lot of data from a similar distribution. Specifically, BridgePure can almost fully restore the dataset availability by using a limited amount of protection leakage, e.g., bringing the accuracy of a trained model back to the level before protection. Moreover, compared to other purification methods based on "noise-adding and denoising" diffusion models, BridgePure avoids detail blurring, artificial distortions or artifacts, and preserves the brushstrokes in the artwork. This demonstrates a critical vulnerability of black-box data protection. Furthermore, we discuss possible mitigation strategies in Appendix D.1 and advocate for considering this type of risk when developing data protection applications.

In summary, our contributions are three-fold:

- We reveal the possible threat of protection leakage against black-box data protection methods;
- We propose *BridgePure* by utilizing DDBM as a powerful purification algorithm that is able to exploit a small amount of protection leakage;
- We conduct comprehensive experiments on purifying existing data protection methods for both classification and generation tasks, where BridgePure consistently outperforms baseline methods.

## 2 Background and Related Work

In this section, we (1) introduce the goals and existing works of data protection on classification models and generative models; (2) outline existing countermeasures that may render the protections ineffective; (3) introduce diffusion bridge models, the key technique we will build on.

### 2.1 Data Protection

Data protection in machine learning aims to achieve two goals: (1) Modify a raw dataset such that *it has low value to machine learning algorithms*; (2) Maintain *usability* for humans, such as publication purposes. We focus on data protection for images.

Formally, we denote the original dataset or *pre-protection* dataset as $\mathcal{D}$, and the *protected dataset* as $\mathcal{D}'$. We refer to the mapping from $\mathcal{D}$ to $\mathcal{D}'$ as *data protection mechanism* $\mathcal{P}$ (e.g., an algorithm),

where $\mathcal{P}$ is applied to every entry in the dataset:

$$\mathcal{P} : \mathcal{D} \to \mathcal{D}', \mathbf{x} \mapsto \mathbf{x}'.$$

To preserve the visual semantics (thus preserving usability for humans), the mechanism $\mathcal{P}$ usually prevents modification from excessively degrading image quality, often relying on an $L_p$-norm constraint on the modification: $\|\mathbf{x}' - \mathbf{x}\|_p \leq \varepsilon$, for some small perturbation budget $\varepsilon > 0$.

Let $\mathcal{M}$ be a training algorithm for a target task and $\mathcal{M}(\mathcal{D}')$ be a model trained using the protected dataset $\mathcal{D}'$. The protection mechanism $\mathcal{P}$ is successful if $\mathcal{M}(\mathcal{D}')$ has degraded performance for the target task. In this paper, we consider two tasks: classification and style mimicry, and their corresponding protection.

**Availability attacks.** Availability attacks[0] can be regarded as a special case of data poisoning attacks. In the context of classification tasks, availability attacks subtly modify the original data, rendering the resulting model $\mathcal{M}$ unusable by reducing its test accuracy to an unacceptable level. Thus, the protected data are often referred to as "unlearnable examples" [e.g., 24].

Over the past few years, this field has advanced rapidly, demonstrating three key trends: (1) Improved performance. Recent techniques can reduce model availability to levels even lower than random guessing [5, 14]. (2) Enhanced resilience. Availability attacks can be effective against both supervised and contrastive learning [19, 48, 65]. Furthermore, robust unlearnable examples have been introduced to counteract weakened protections caused by adversarial training [12, 15, 66]. (3) Transferable protection. Recent methods leverage image concepts and semantics to generate protective perturbations, enabling cross-dataset protection [4, 74]. This remarkable progress highlights the potential of availability attacks as a practical data protection strategy in real-world applications.

**Style mimicry protections.** Consider an artist with artwork $\mathcal{D}$ in a distinctive style $\mathcal{S}$. Latent diffusion models (LDMs) [50] can readily fine-tune on $\mathcal{D}$ to generate new images mimicking style $\mathcal{S}$ from text prompts. To prevent such unauthorized style replication, data protection mechanisms $\mathcal{P}$ modify the latent representation of $\mathcal{D}$ to align with a different public dataset, making style extraction through LDM fine-tuning ineffective. Our analysis focuses on two recent methods: Glaze [56] and Mist [32], which prevent mimicry by applying imperceptible protective modifications to paintings.

## 2.2 Circumventing Data Protection

To understand the real effectiveness of data protection, existing approaches propose techniques that degrade data protection. Specifically:

**Purification-based methods.** Adversarial purification was first introduced to sanitize adversarial examples at test time [52, 59, 70]. DiffPure [42] employs pre-trained diffusion models to remove undesired noise from the perturbed images. In the context of protection removal for classification tasks, AVATAR [9] borrows a diffusion model pre-trained on the unprotected dataset to purify the protected dataset. LE-JCDP [25] fine-tunes a pre-trained diffusion model on additional data (i.e., the test set) and regularizes the sampling stage to improve the quality of purified images. D-VAE [72] leverages a variational auto-encoder-based method to disentangle protective perturbations from protected images, which requires no additional data. Regarding style mimicry tasks, DiffPure, IMPRESS [3], Noisy Upscaling [21, 41], GrIDPure [75], and PDM [68] prove effective in undermining the protection provided by current popular tools [21].

**Other methods[1].** The imperceptible nature of protective modifications enables adversarial training to mitigate the protection efficacy for classification tasks [39, 62]. Additionally, processing the protected images by traditional and specially picked data augmentations can restore availability to some extent [34, 45, 78].

Although existing methods often rely on pretrained models or require training models from scratch with a large amount of protected data, they still leave an availability gap between the purified and

---

[0]Note that while "availability attack" here refers to data protection methods, it can also mean indiscriminate data poisoning attacks. See Appendix A for a complete discussion.

[1]We discuss the line of work that shows a "false sense of security" in current data protection in Appendix A.

original datasets. In this work, we show that under a novel yet realistic threat model of limited protection leakage, the strength of data protection can be almost completely diminished.

## 2.3 Diffusion Bridge Models

Denote by $q_{\text{data}}(\mathbf{x})$ the initial data distribution. We construct a diffusion process with a set of time-indexed variables $\{\mathbf{x}_t\}_{t=0}^T$. Diffusion models transporting the initial distribution to a standard Gaussian distribution are associated with the following SDE [60]:

$$d\mathbf{x}_t = \mathbf{f}(\mathbf{x}_t, t)\, dt + g(t)\, d\mathbf{w}_t, \quad \mathbf{x}_0 \sim q_{\text{data}}(\mathbf{x}), \tag{1}$$

where $\mathbf{f} : \mathbb{R}^d \times [0, T] \to \mathbb{R}^d$ is vector-valued drift function, $g : [0, T] \to \mathbb{R}$ is a scalar-valued diffusion coefficient and $\mathbf{w}_t$ is a Wiener process.

We are interested in the transportation between two arbitrary data distributions. Assume the diffusion process $\{\mathbf{x}_t\}_{t=0}^T$ satisfies $\mathbf{x}_0 \sim q_{\text{data}}(\mathbf{x})$ and $\mathbf{x}_T = \mathbf{x}'$ as a fixed endpoint. This process can be modeled as the solution of the following SDE [10, 49]:

$$d\mathbf{x}_t = [\mathbf{f}(\mathbf{x}_t, t) + g^2(t)\mathbf{h}(\mathbf{x}_t, t, \mathbf{x}', T)]\, dt + g(t)\, d\mathbf{w}_t, \mathbf{x}_0 \sim q_{\text{data}}(\mathbf{x}), \mathbf{x}_T = \mathbf{x}', \tag{2}$$

where $\mathbf{h}(x, t, \mathbf{x}', T) = \nabla_{\mathbf{x}_t} \log p(\mathbf{x}_T | \mathbf{x}_t)|_{\mathbf{x}_t = x, \mathbf{x}_T = \mathbf{x}'}$ is the gradient of the log transition kernel from $t$ to $T$ generated by the original SDE (1). One can reverse the process (2) as follows Zhou et al. [76]:

$$d\mathbf{x}_t = [\mathbf{f}(\mathbf{x}_t, t) - g^2(t)(\mathbf{s}(\mathbf{x}_t, t, \mathbf{x}', T) - \mathbf{h}(\mathbf{x}_t, t, \mathbf{x}', T))]\, dt + g(t)\, d\hat{\mathbf{w}}_t, \quad \mathbf{x}_T = \mathbf{x}', \tag{3}$$

where $\hat{\mathbf{w}}_t$ is a reverse Wiener process, $q$ is the transition kernel of (2), and the score function $\mathbf{s}(\mathbf{x}, t, \mathbf{x}', T) = \nabla_{\mathbf{x}_t} \log q(\mathbf{x}_t | \mathbf{x}_T)|_{\mathbf{x}_t = \mathbf{x}, \mathbf{x}_T = \mathbf{x}'}$. The time-reversed SDE (3) is known to be associated with a probability flow ODE [60]:

$$d\mathbf{x}_t = [\mathbf{f}(\mathbf{x}_t, t) - g^2(t)(\tfrac{1}{2}\mathbf{s}(\mathbf{x}_t, t, \mathbf{x}', T) - \mathbf{h}(\mathbf{x}_t, t, \mathbf{x}', T))]\, dt. \tag{4}$$

Accordingly, a denoising diffusion bridge model (DDBM) parametrized by $\theta$ is trained by minimizing the following (denoising) score matching objective:

$$\mathcal{L}(\theta) = \mathbb{E}_{\mathbf{x}_t, \mathbf{x}_0, \mathbf{x}_T, t}[\lambda(t)\|\mathbf{s}_\theta(\mathbf{x}_t, \mathbf{x}_T, t) - \nabla_{\mathbf{x}_t} \log q(\mathbf{x}_t | \mathbf{x}_0, \mathbf{x}_T)\|^2] \tag{5}$$

where $(\mathbf{x}_0, \mathbf{x}_T) \sim q_{\text{data}}(\mathbf{x}, \mathbf{x}')$, $\mathbf{x}_t \sim q(\mathbf{x}_t | \mathbf{x}_0, \mathbf{x}_T)$ and $\lambda(t)$ is the weighting coefficient.

## 3 Threat Model

In this section, we introduce (1) how data protection provides service for individual data owners; (2) possible loopholes and an attack pathway; (3) the notion of protection leakage, and (4) differences with existing works. Figure 1 summarizes the threat model considered in this paper.

**Protection service.** To leverage availability attacks for data protection, a black-box service can be offered to data owners without requiring machine learning expertise. For instance, Glaze [56] provides a user-friendly application where individuals can locally apply the protection mechanism $\mathcal{P}$ to their personal dataset $\mathcal{D}$, generating a protected version $\mathcal{D}'$. In our work,

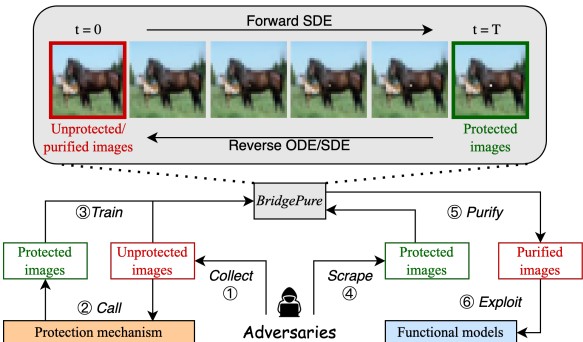

Figure 1: The threat model and illustration of BridgePure. Sequential images show the ODE sampling (purification) process of an example image protected by One-Pixel Shortcut [67].

we assume all attacks operate in a black-box manner, meaning both data owners and adversaries have no knowledge of $\mathcal{P}$'s internal mechanisms.

**Adversary.** Note that while such black-box services are convenient for data owners, they are accessible to *anyone*, without any ownership verification. This means adversaries can potentially use these services to generate protected versions of data belonging to others. For instance, if there exist publicly available unprotected images belonging to a data owner, an adversary $\mathcal{A}$ might use these unprotected images to form an additional dataset $\mathcal{D}_a$. Note that $\mathcal{D}_a$ must be drawn from the same distribution as $\mathcal{D}$, or from a sufficiently similar distribution.

Formally, we define the adversary's capabilities as: (1) Access to a large dataset $\mathcal{D}'$ of protected data; (2) Access to a small additional dataset $\mathcal{D}_a$ of unprotected data, where $|\mathcal{D}_a| \ll |\mathcal{D}'|$ and $\mathcal{D}_a \cap \mathcal{D} = \emptyset$ (with $\mathcal{D}$ being the original unprotected dataset corresponding to $\mathcal{D}'$); (3) Access to the black-box protection mechanism $\mathcal{P}$.

**Protection leakage.** By querying the protection mechanism $\mathcal{P}$ on the collected dataset $\mathcal{D}_a$, the adversary $\mathcal{A}$ obtains a paired dataset $\widehat{\mathcal{D}}_a := \{(\mathbf{x}, \mathcal{P}(\mathbf{x})) | \mathbf{x} \in \mathcal{D}_a\}$, containing both unprotected and protected versions of each data point. While $\mathcal{P}$ remains black-box to $\mathcal{A}$, this paired dataset $\widehat{\mathcal{D}}_a$ reveals information about $\mathcal{P}$. For real-world applications of data protection, a critical question emerges: *Does the information leaked through $\widehat{\mathcal{D}}_a$ compromise the protection provided by $\mathcal{P}$?*

Our main finding reveals that protection leakage enables the construction of a powerful purification mechanism $\mathcal{P}^{-1}$ that approximately reverses the protection $\mathcal{P}$. Using this mechanism, an adversary $\mathcal{A}$ can purify the protected dataset $\mathcal{D}'$ to obtain $\mathcal{P}^{-1}(\mathcal{D}')$, which closely matches the availability of the original dataset $\mathcal{D}$.

**Difference with other purification methods.** Notably, compared to existing circumvention methods discussed in Section 2.2, our approach is distinctive in two ways: (1) Our threat model assumes access to the black-box mechanism $\mathcal{P}$, providing the adversary greater (but viable) capabilities; (2) Our method requires only limited unprotected samples to develop a purification from scratch, unlike DiffPure [42] and AVATAR [9], for which models are pre-trained using enormous additional data.

We argue that even with a small amount of unprotected data, attackers can bypass existing data protection mechanisms using moderate means—without requiring pre-trained models on specific types of data or massive computational resources. It also confirms that protection has a time and space dimension: any information that has ever been leaked or will be leaked in the future is significantly harder to protect. Similarly, protecting data in only one place is far from sufficient (for example, securing online data while neglecting offline data).

# 4 Bridge Purification

In this section, we specify the possible impact of protection leakage by introducing Bridge Purification (*BridgePure*), a method that learns the inverse protection mechanism $\mathcal{P}^{-1}$ from limited protection leakage $\widehat{\mathcal{D}}_a = \{(\mathbf{x}, \mathbf{x}') | \mathbf{x} \in \mathcal{D}_a, \mathbf{x}' = \mathcal{P}(\mathbf{x})\}$, where each pair contains unprotected and protected versions of the same data. BridgePure works by modeling and then inverting the transformation between the original and protected data.

**Bridge training.** Assume the pairs $(\mathbf{x}, \mathbf{x}')$ come from a joint distribution $q_{\text{data}}(\mathbf{x}, \mathbf{x}')$, where $\mathbf{x}' = \mathcal{P}(\mathbf{x})$. We aim to learn $\mathcal{P}^{-1}$ that approximately samples from $q_{\text{data}}(\mathbf{x}|\mathbf{x}')$, i.e., purifying the protected data $\mathbf{x}'$. We first construct the stochastic process $\{\mathbf{x}_t\}_{t=0}^{T}$ that starts from $\mathbf{x}_0 = \mathbf{x}$ and ends at $\mathbf{x}_T = \mathbf{x}'$, where $q(\mathbf{x}_0, \mathbf{x}_T)$ approximates the true distribution $q_{\text{data}}(\mathbf{x}, \mathbf{x}')$. This process can be modeled by SDE (2) in Section 2.3. We can reverse the process using the SDE (3) and ODE (4). Given the protection leakage $\widehat{\mathcal{D}}_a$, we train a denoising diffusion bridge model [76] *from scratch* via minimizing the score-matching loss in eq. (5) on $\widehat{\mathcal{D}}_a$.

**Sampling and purification.** Different from standard diffusion models which perform unconditional sampling, BridgePure's sampling process requires each step to be conditioned on the endpoint $\mathbf{x}'$ (i.e., the protected data). Following Zhou et al. [76], we deploy a hybrid sampling approach that combines Euler-Maruyama and Heun sampling methods, with a hyperparameter $s \in [0, 1]$ controlling the sampling randomness. When $s = 0$, the sampling is deterministic, and higher values of $s$ introduce greater randomness. Choosing an appropriate $s$ can enhance sampling quality and improve purified datasets' availability, which we analyze through ablation studies on $s$ in Section 5.4.

BridgePure purifies the protected dataset $\mathcal{D}'$ by performing conditional sampling for each protected sample $\mathbf{x}'$. As shown in Figure 1, the purification process gradually removes protective features, such as the white spot on the horse's chest. After obtaining the purified dataset $\mathcal{P}^{-1}(\mathcal{D}')$, we evaluate purification effectiveness through model performance on the purified data, denoted as $\mathcal{M}(\mathcal{P}^{-1}(\mathcal{D}'))$.

**Pre-processing.** When $\widehat{\mathcal{D}}_a$ contains a small number of leaked pairs, BridgePure may overfit to the limited data and fail to generalize well to the protected dataset $\mathcal{D}'$. To address this limitation, we introduce Gaussian noise to the protected data, inspired by the diffusion process:

$$\mathcal{G}_\beta(\mathbf{x}') = \sqrt{1-\beta}\mathbf{x}' + \sqrt{\beta}\mathbf{z}, \quad \mathbf{z} \sim \mathcal{N}(\mathbf{0}; \mathbf{I}).$$

After pre-processing, the protection leakage becomes $\widehat{\mathcal{D}}_a = \{(\mathbf{x}, \mathcal{G}_\beta(\mathbf{x}')) | \mathbf{x} \in \mathcal{D}_a, \mathbf{x}' = \mathcal{P}(\mathbf{x})\}$ and the protected dataset is $\mathcal{D}' = \{\mathcal{G}_\beta(\mathbf{x}') | \mathbf{x} \in \mathcal{D}, \mathbf{x}' = \mathcal{P}(\mathbf{x})\}$. BridgePure learns to model the transformation between $\mathbf{x}$ and $\mathcal{G}_\beta(\mathbf{x}')$ using $\widehat{\mathcal{D}}_a$, then purifies $\mathcal{G}_\beta(\mathbf{x}') \in \mathcal{D}'$ by sampling approximately from $q_{\text{data}}(\mathbf{x}|\mathcal{G}_\beta(\mathbf{x}'))$. The effectiveness of BridgePure can be enhanced through the appropriate selection of the pre-processing parameter $\beta$, which we examine through ablation studies in Section 5.4.

Table 1: Purification performance on CIFAR-10 and CIFAR-100 against nine availability attacks. The best restoration results are emphasized in **bold**. We underline to denote the least number of pairs required for BridgePure to surpass other baseline methods. We run five random trials for evaluation and report the mean value and standard deviation.

| | AR | DC | EM | GUE | LSP | NTGA | OPS | REM | TAP |
|---|---|---|---|---|---|---|---|---|---|
| | CIFAR-10 (94.01±0.15) | | | | | | | | |
| Protected | 13.52±0.63 | 15.10±0.81 | 23.79±0.13 | 12.76±0.44 | 13.85±0.96 | 12.87±0.23 | 13.67±1.80 | 20.96±1.70 | 9.51±0.67 |
| PGD-AT | 81.78±0.31 | 82.56±0.23 | 83.86±0.06 | 83.80±0.28 | 83.46±0.09 | 83.39±0.22 | 9.60±1.58 | 85.47±0.17 | 81.82±0.12 |
| D-VAE | 90.22±0.44 | 88.63±0.28 | 88.75±0.22 | 89.80±0.43 | 90.04±0.22 | 87.88±0.25 | 89.48±0.37 | 83.07±0.38 | 83.22±0.49 |
| AVATAR | 91.41±0.13 | 89.04±0.17 | 88.46±0.24 | 88.05±0.31 | 89.05±0.29 | 88.50±0.30 | 87.87±0.19 | 89.66±0.47 | 90.76±0.24 |
| LE-JCDP | 92.07±0.21 | 91.63±0.23 | 90.69±0.31 | 90.79±0.20 | 91.22±0.31 | 91.57±0.25 | 58.60±1.28 | 90.39±0.24 | 91.60±0.14 |
| BridgePure-0.5K | **93.86**±0.27 | 93.76±0.17 | 93.64±0.22 | 93.70±0.11 | 93.76±0.18 | **94.07**±0.18 | 93.31±0.19 | 84.34±0.52 | 86.81±0.31 |
| BridgePure-1K | 92.48±0.11 | 93.78±0.25 | 93.73±0.15 | 93.80±0.20 | 93.84±0.19 | 93.94±0.08 | 93.49±0.26 | 92.69±0.25 | 87.62±0.05 |
| BridgePure-2K | 93.84±0.22 | **93.93**±0.20 | 93.81±0.22 | **93.97**±0.15 | **93.99**±0.34 | 94.00±0.16 | 93.31±0.36 | 93.49±0.18 | 88.60±0.22 |
| BridgePure-4K | 93.56±0.21 | 93.81±0.05 | **93.87**±0.15 | 93.84±0.21 | 93.93±0.27 | 93.93±0.12 | **93.50**±0.28 | **93.50**±0.11 | **92.91**±0.12 |
| | CIFAR-100 (74.27±0.45) | | | | | | | | |
| Protected | 2.02±0.12 | 36.10±0.67 | 6.73±0.12 | 19.50±0.48 | 2.56±0.16 | 1.51±0.22 | 12.18±0.52 | 7.07±0.19 | 3.59±0.12 |
| PGD-AT | 56.37±0.25 | 55.21±0.40 | 56.25±0.29 | 57.38±0.27 | 56.19±0.28 | 54.77±0.25 | 7.59±0.32 | 56.81±0.19 | 54.59±0.28 |
| D-VAE | 62.14±0.32 | 55.91±0.92 | 60.25±0.25 | 60.79±0.62 | 61.36±0.75 | 59.34±0.64 | 62.83±0.67 | 63.06±0.31 | 53.82±0.91 |
| AVATAR | 65.45±0.32 | 63.48±0.26 | 62.77±0.56 | 62.10±0.22 | 62.95±0.38 | 62.60±0.22 | 60.68±0.56 | 65.36±0.38 | 64.50±0.23 |
| LE-JCDP | 69.15±0.22 | 68.49±0.42 | 67.76±0.31 | 67.36±0.42 | 68.23±0.40 | 68.35±0.19 | 39.10±0.40 | 68.76±0.23 | 68.39±0.39 |
| BridgePure-0.5K | 67.49±0.31 | 73.69±0.21 | 73.17±0.13 | 72.69±0.49 | 73.33±0.77 | 69.11±0.86 | **74.18**±0.31 | 66.53±0.29 | 62.75±0.25 |
| BridgePure-1K | 68.63±0.84 | 73.62±0.34 | 73.31±0.42 | 72.92±0.62 | 73.93±0.24 | 69.96±0.47 | 74.22±0.30 | 66.30±0.36 | 62.58±0.28 |
| BridgePure-2K | 68.05±0.16 | 73.83±0.15 | **73.70**±0.30 | 73.55±0.29 | 73.86±0.56 | 73.90±0.19 | 73.96±0.40 | 72.38±0.44 | 64.96±0.27 |
| BridgePure-4K | **72.44**±0.47 | **73.97**±0.18 | 73.52±0.57 | **73.92**±0.09 | **74.56**±0.40 | **74.23**±0.23 | **74.18**±0.38 | **72.95**±0.10 | **70.96**±0.15 |

# 5 Experiments

In this section, we (1) introduce our experimental setting, (2) present BridgePure's purification results on purifying availability attacks and style mimicry protection, and (3) conduct ablation studies.

## 5.1 Experimental Setting

**Datasets.** Our classification experiments use CIFAR-10/100 [29], ImageNet-Subset,[2] WebFace-Subset,[3] Cars [28], and Pets [43] datasets. For style mimicry experiments, we use artwork from artist @*nulevoy*,[3] with details provided in Section 5.3.

**Protections.** On classification tasks, we leverage 14 availability attacks to simulate different data protection tools. Among them, AR [54] and LSP [71] are $L_2$-norm attacks, OPS [67] is an $L_0$-norm attack, while the rest are $L_\infty$-norm attacks including DC [13], EM [24], GUE [33], NTGA [73],

---

[2]ImageNet-Subset is a subset of ImageNet [8] containing 100 classes. WebFace-Subset is a subset of CASIA-WebFace [69] containing 100 identities. See Appendix B.1 for detailed settings.

[3]https://www.artstation.com/nulevoy, usage with consent from the artist.

REM [15], TAP [14], CP [19], TUE [48], AUE [65], UC and UC-CLIP [74]. If not otherwise stated, these $L_\infty$-norm attacks use a modification budget $\varepsilon = 8/255$. More details about protection generation are available in Appendix B.2. On generation tasks, we deploy two style mimicry protection tools, i.e., Glaze v2.1 [56] and Mist [32].

**BridgePure.** We train BridgePure using a small set of (unprotected, protected) pairs to purify large-scale protected data and evaluate the purified dataset's availability. We denote BridgePure-$N$ as the model trained on $N$ pairs, ensuring these training pairs are distinct from the protected samples to be purified. Following Section 4, we apply Gaussian perturbation with parameter $\beta$ during preprocessing and control sampling randomness via parameter $s$. For CIFAR-10 and CIFAR-100, we report BridgePure's best performance across four configurations: $s \in \{0.33, 0.8\}$ and $\beta \in \{0, 0.02\}$. For ImageNet-Subset, WebFace-Subset, Cars, and Pets, we report results with $s \in \{0.33, 0.8\}$ and $\beta = 0$. For style mimicry protection, we set $s = \beta = 0$.

**Purification baselines.** We compare BridgePure with existing purification-based methods in Section 2.2, including adversarial training [39] and three purification baselines, including D-VAE [72], AVATAR [9], and LE-JCDP [25] on CIFAR-10 and CIFAR-100. Notably, D-VAE requires no additional data, while AVATAR uses a diffusion model trained on the unprotected dataset containing 50K images, and LE-JCDP fine-tunes a diffusion model on the unprotected dataset containing 10K images. BridgePure leverages a significantly smaller amount of protection leakage for training—ranging from only 0.5K to 4K pairs. For ImageNet-Subset and WebFace-Subset comparisons with DiffPure [42], details are provided in the relevant section.

## 5.2 Purifying Availability Attacks

**Main results.** We evaluate four levels of protection leakage: $N = 500, 1000, 2000,$ and $4000$ pairs of unprotected and protected images. For each level, an adversary trains a BridgePure model to attempt purification of the protected dataset. In Table 1, we compare BridgePure with four baseline methods: adversarial training using PGD-10 with budget $8/255$ in $L_\infty$-norm, D-VAE, AVATAR, and LE-JCDP. The results demonstrate the significant impact of protection leakage in three aspects: (1) *Restoration with limited leakage*: BridgePure substantially restores dataset availability even with a few leaked pairs. (2) *Superior performance with higher budgets*: Using up to 4K pairs, BridgePure consistently outperforms all baseline methods across nine attacks. (3) *Closing the availability gap*: BridgePure's protection-specific design increasingly eliminates the availability gap, approaching perfect restoration as protection leakage increases.

Figure 2: Performance comparison with augmentation-based methods, and protection dilution on CIFAR-100.

Table 2: Purification performance on ImageNet-Subset and WebFace-Subset against three availability attacks.

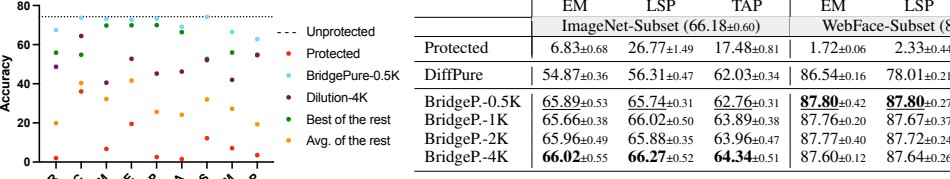

| | EM | LSP | TAP | EM | LSP | TAP |
|---|---|---|---|---|---|---|
| | ImageNet-Subset (66.18±0.60) | | | WebFace-Subset (87.84±0.27) | | |
| Protected | 6.83±0.68 | 26.77±1.49 | 17.48±0.81 | 1.72±0.06 | 2.33±0.44 | 3.24±0.52 |
| DiffPure | 54.87±0.36 | 56.31±0.47 | 62.03±0.34 | 86.54±0.16 | 78.01±0.21 | 79.59±0.79 |
| BridgeP.-0.5K | 65.89±0.53 | 65.74±0.31 | 62.76±0.31 | **87.80**±0.42 | **87.80**±0.27 | 82.48±0.23 |
| BridgeP.-1K | 65.66±0.38 | 66.02±0.50 | 63.89±0.38 | 87.76±0.20 | 87.67±0.37 | 86.38±0.26 |
| BridgeP.-2K | 65.96±0.49 | 65.88±0.35 | 63.96±0.47 | 87.77±0.40 | 87.72±0.24 | 87.27±0.42 |
| BridgeP.-4K | **66.02**±0.55 | **66.27**±0.52 | **64.34**±0.51 | 87.60±0.12 | 87.64±0.26 | **87.46**±0.19 |

Moreover, Figures 2 and 14 demonstrate that BridgePure consistently outperforms eight augmentation-based circumvention methods. (See Appendix C.6 for a detailed illustration of this comparison.). We also considered the scenario where the adversary dilutes the protected dataset with a sufficiently large amount of unprotected data. The results indicate that 500 leaked pairs have a significantly greater destructive impact and harm than 4,000 leaked unprotected samples.

In Table 2, we evaluate BridgePure on ImageNet-Subset and WebFace-Subset to illustrate the risk of protection leakage in real-world scenarios. For baseline DiffPure, the diffusion model for ImageNet-Subset is trained on the entire ImageNet, and that for WebFace-Subset is trained on CelebA [35]. We report the best results of DiffPure among four selections of sampling step, i.e., $t^* \in \{0.1, 0.2, 0.3, 0.4\}$. When the amount of leaked pairs is 500, our BridgePure already surpasses DiffPure on the two datasets. Moreover, BridgePure can restore the availability to the original levels as the leakage grows.

**Label-agnostic case.** We consider label-agnostic variants of availability attacks, i.e., UC and UC-CLIP, whose protection generation depends on clustering in the feature space of a pre-trained encoder such as CLIP [46]. We adopt their default implementation settings where the number of surrogate clusters is 10 and the protection budget is $16/255$ in $L_\infty$ norm. In Table 3, BridgePure with at most 1000 leaked pairs can purify the protected datasets to the original availability levels.

Table 3: Purification performance on Cars and Pets against two label-agnostic availability attacks.

|  | UC | UC-CLIP | UC | UC-CLIP |
|---|---|---|---|---|
|  | Cars (43.25±1.71) | | Pets (49.56±0.81) | |
| Protected | 25.91±4.58 | 10.93±2.78 | 20.91±1.17 | 24.07±4.92 |
| BridgeP.-0.5K | 43.65±1.32 | 42.72±1.64 | 50.03±0.80 | 50.70±1.44 |
| BridgeP.-1K | 42.32±1.25 | 43.45±2.44 | 49.27±3.08 | 49.75±0.78 |

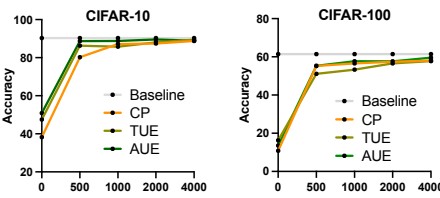

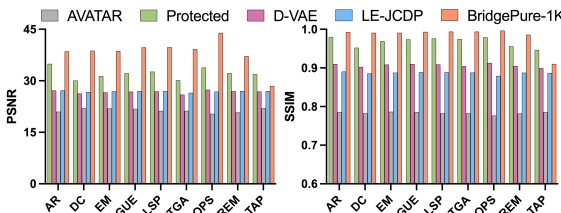

Figure 3: Purification performance against availability attacks that SimCLR evaluates.

Figure 4: PSNR and SSIM between processed datasets and the original CIFAR-10.

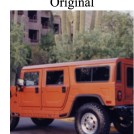 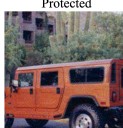 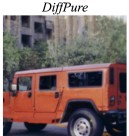 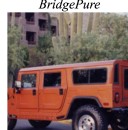 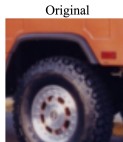 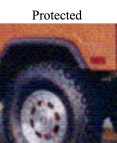 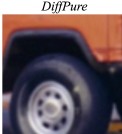 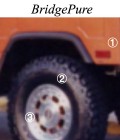

Figure 5: Purification outcomes on UC-protected Cars. The **left** is the overview comparison and the **right** shows local details around the wheel. We point out (1) the light, (2) the tire, and (3) the wheel hub where BridgePure-0.5K preserves the original texture while DiffPure ($t^* = 0.2$) blurs details.

**Contrastive learning case.** We consider availability attacks that transfer to contrastive learning algorithms. We purify CP, TUE, and AUE by BridgePure and then train classifiers using SimCLR [6] and linear probing. Figure 3 shows that limited protection leakage enables BridgePure to recover the availability for contrastive learning significantly.

**Purified image quality.** A distinct feature of BridgePure is its conditional generation based on the protected images. We observe that this approach enables high-quality restoration, preserves image details, and avoids artificial distortions or artifacts. Specifically, in Figure 4, we evaluate the similarity between the original (unprotected) data and their purified versions with PSNR and SSIM metrics. We also present the similarity between the protected and unprotected pairs as a baseline. We observe that our method outperforms all baseline purification methods in terms of restoring the unprotected data. Moreover, our method consistently improves image similarity through purification, while other methods downgrade the similarity compared with the protected baseline.

Moreover, in Figure 5, we compare the details of the purified images generated by DiffPure and BridgePure. In terms of the purification mechanism, DiffPure adds Gaussian noise to protected images and aligns them with learned trajectories before reverse sampling. We observe that such an unconditional process could cause the loss of texture details. In contrast, BridgePure's conditional sampling preserves fine-grained features. Concretely, details of the vehicle purified by BridgePure, such as lights, tires, and wheel hubs, are in sharper clarity than those purified by DiffPure.

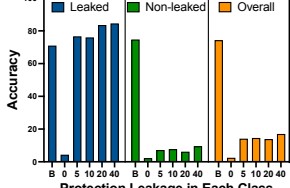 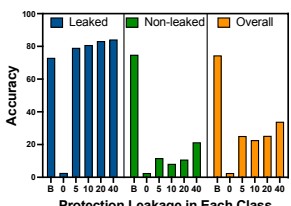

Figure 6: Performance with partial protection leakage within 10 classes (**left**) and 20 classes (**right**) of LSP-protected CIFAR-100. The **x-axis** represents the number of leaked pairs in each leaked class and "B" stands for the unprotected baseline. Here $s = 0.33$ and $\beta = 0$.

**Partial protection leakage.** The adversary is often limited by cost and capability, making it impossible to collect all types of clean images, or they may only be interested in specific categories within the dataset. Thus, we consider a scenario where the adversary aims to purify protected images from certain classes rather than the whole protected dataset $\mathcal{D}'$. In Figure 6, we purify LSP-protected CIFAR-100 using partial protection leakage within 10/20 random classes and report the accuracy of leaked, non-leaked, and all classes, respectively. The results demonstrate that partial protection leakage poses an even more significant risk to relevant classes. For example, 5 pairs from each class are sufficient to make the test accuracy of the target classes better than the unprotected baseline, and more pairs will improve it further.

**Mixture of protection.** The mechanism $\mathcal{P}$ could possibly employ multiple availability attacks to protect data. In such cases, the protection leakage also contains a mixture of differently protected pairs. In Table 4, we consider a scenario in which $\mathcal{P}$ randomly applies one of five attacks to a given input data. We observe that, firstly, the mixture of protection harms the protection performance and this approach is not desirable; secondly, BridgePure is still very effective in restoring availability when the leakage amount is relatively small.

Table 4: Purification performance in the mixed-attacks scenario, where five availability attacks including AR, EM, LSP, OPS, and TAP are randomly applied.

|  | Protected | BridgePure | | |
|---|---|---|---|---|
|  |  | 0.25K | 0.5K | 1K |
| CIFAR-10 (94.01) | $61.60_{\pm1.78}$ | $93.00_{\pm0.26}$ | $93.14_{\pm0.24}$ | $93.01_{\pm0.20}$ |
| CIFAR-100 (74.27) | $51.57_{\pm2.15}$ | $71.31_{\pm0.50}$ | $72.00_{\pm0.24}$ | $72.77_{\pm0.33}$ |

## 5.3 Purifying Style Mimicry Protection

In this section, we investigate the threat of protection leakage to copyright protection for generative models. We consider art style mimicry on the artwork from an artist *@nulevoy* with consent. We first fine-tune Stable Diffusion v2.1 [50] using 20 captioned paintings following the implementation of Hönig et al. [21]. We then reproduce the style of the artist with a list of prompts during inference. Our implementation details are available in Appendix B.5.

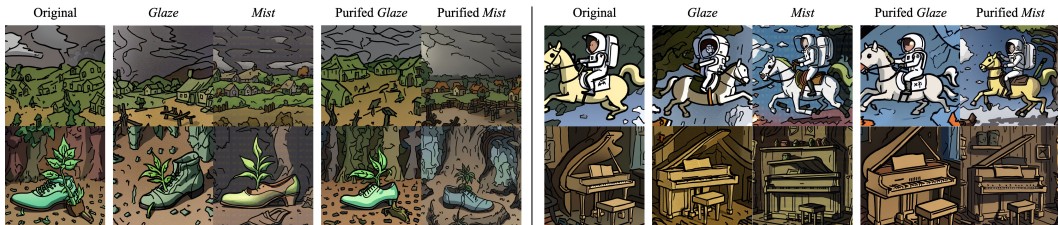

Figure 7: Purification performance of BridgePure-5 (**top**) and BridgePure-10 (**bottom**) for style mimicry. The presented paintings are mimicry outcomes of fine-tuned generative models.

For style mimicry protection, we apply Glaze and Mist to protect the 20 paintings we used previously. We assume protection leakage of only 5 or 10 unprotected paintings of the same artist and call these public protection tools to obtain (unprotected, protected) pairs for BridgePure training. Finally, the 20 protected paintings are purified by BridgePure and fed into the style mimicry pipeline.

Figures 7 and 10 show the style mimicry outcomes given different text prompts. Models fine-tuned on Glaze-protected artwork produce images filled with irregular patterns, while artwork protected by Mist leads fine-tuned models to generate artistic works with regular block-like perturbations. After purification by BridgePure, images protected by Glaze and Mist can no longer cause fine-tuned models to generate artwork with protective cloaks. Our results again suggest that for style mimicry, protection leakage poses a strong threat to existing data protection tools.

Due to page limitations in the main text, we will compare our BridgePure and other advanced approaches, including GrIDPure [75], PDM [68], and NoisyUpscaling [21], for purifying protected paintings in Appendix C.4. As shown in Figures 11 and 12, there BridgePure effectively removes

protective perturbations while preserving the intricate details of the painting—an achievement that other approaches fall short of.

## 5.4 Ablation Study

Figure 8 shows that pre-processing with Gaussian noise can improve the availability restoration against some availability attacks which are "harder" to purify, e.g., TAP. However, it also presents a performance ceiling for other protections, e.g., EM and LSP, and harms their purification results. Regarding the sampling randomness, while larger randomness slightly reduces the accuracy for some protections, e.g., EM, LSP, and OPS, it can largely benefit the purification against TAP, REM, and AR.

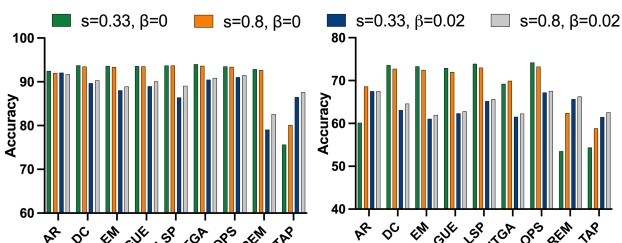

Figure 8: Influence of $s$ and $\beta$ on BridgePure-1K performance on CIFAR-10 (**left**) and CIFAR-100 (**right**).

In summary, different protection methods are subject to different choices of the optimal hyperparameter. Our results in this section reveal the worst-case damage caused by protection leakage by reporting the best-performing BridgePure within a limited number of trials.

# 6 Potential Countermeasures

To mitigate protection leakage and defend against BridgePure, both service providers and users can adopt proactive countermeasures. For service providers, the critical safeguard is to limit their ability to use the protection mechanism to generate the corresponding protected versions. Potential countermeasures include limiting the reproducibility of protection, avoiding offline deployment, and verifying user identity and data ownership. An alternative line of defense against the BridgePure threat is to design protection mechanisms that are resilient to its purification capabilities.

For protection service users, the most straightforward defense is to minimize the exposure of unprotected images and rely on trustworthy protection service providers. Another promising strategy is to design preprocessing techniques that work jointly with the protection mechanism to enhance resistance against BridgePure. In addition, users may strategically release crafted decoy images to contaminate an attacker's training data, thereby diminishing BridgePure's effectiveness on their genuine protected content.

The effectiveness of these mitigation strategies remains to be thoroughly evaluated, and we plan to explore them further in future work. A more detailed discussion of potential countermeasures is provided in Appendix D.1.

# 7 Conclusion

In this paper, we identify a critical vulnerability in black-box data protection systems: *protection leakage*. We demonstrate that using a small number of leaked pairs, an adversary can train a diffusion bridge model, *BridgePure*, to effectively circumvent the protection mechanism. Our empirical results show that under this threat model, *BridgePure* exposes fundamental vulnerabilities in current data protection systems for both classification and generation tasks.

**Limitations and future work.**   Our findings underscore the urgent need to address protection leakage. For protection service providers, it is essential to develop both system-level and algorithm-level countermeasures to mitigate the threat posed by BridgePure. We have already notified several blackbox protection service providers of these risks and shared our recommendations. Furthermore, exploring user-oriented defense strategies represents another promising direction for future research.

In addition, the data protection methods discussed in this paper are limited to the image domain. Investigating the reliability of data protection approaches in other domains is another important direction for future work.

## Acknowledgment

We sincerely thank Stanislav Voloshin (@*nulevoy*) for his permission to present experimental results based on his artwork in our paper. GK and YY gratefully acknowledge NSERC, the Canada CIFAR AI Chairs program and the Ontario Early Researcher program for funding support. XSG is supported by the Strategic Priority Research Program of CAS Grant XDA0480502, Robotic AI-Scientist Platform of CAS, and NSFC Grants 12288201 and 92270001.

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

# A  Data Protection and Data Poisoning Attacks

In this section, we formalize the relationship between data protection and data poisoning attacks. First, let us define data poisoning attacks: given a clean training set $\mathcal{D}_c$, data poisoning attacks create an additional poisoned set $\mathcal{D}_p$ such that a model trained on $\mathcal{D}_c \cup \mathcal{D}_p$ exhibits behavior aligned with the adversary's objective. These attacks can be categorized as: availability (or indiscriminate) attacks [e.g., 2, 26, 27, 36, 37, 38, 40, 61] that reduce overall test performance, targeted attacks [e.g., 1, 16, 18, 55, 77], or backdoor attacks [e.g., 7, 17, 51, 64] that compromise model integrity for specific test samples or trigger patterns.

Data protection can be viewed as a special case of availability attacks where: (1) $|\mathcal{D}_c| = 0$, (2) $\mathcal{D}_p$ is the protected dataset $\mathcal{D}'$, and (3) the adversary role is taken by the data protection service provider.

Finally, the inadequacy of data poisoning as a protection mechanism has been conclusively demonstrated, both through conceptual analysis [47] and technical evaluation [21, 44]. Radiya-Dixit et al. [47] identify a fundamental limitation in data protection methods: their "once for all" deployment mechanism fails to protect historical data and lacks cross-model transferability. While recent advances in transferable availability attacks [19, 48, 65] have partially addressed the model transferability challenge, our work reveals that the vulnerability of historical unprotected data (protection leakage) poses an even more significant security risk.

# B  Experiment Settings

## B.1  Datasets

**CIFAR-10/100.**  For CIFAR-10 and CIFAR-100 [30], the training set is divided into two parts: a set to be protected which contains 40,000 images, and a reference set comprising the remaining data. The images are 32×32 pixels.

**ImageNet-Subset.**  The ImageNet-100 dataset consists of 100 classes selected from the full ImageNet dataset [8]. Following Fu et al. [15], Huang et al. [24], and Qin et al. [45], we use a subset of ImageNet-100 containing 85,000 images. The test set includes 50,000 images, the set to be protected contains 25,000 images, and the reference set includes 10,000 images. Images in both the protection and reference sets are resized to 224×224 pixels. For test images, the shorter edge is resized to 256 pixels, followed by a center crop to 224×224.

**WebFace-Subset.**  The CASIA-WebFace dataset [69] contains 494,414 face images of 10,575 real identities. We select the top 100 identities with the most images, resulting in a dataset of 44,697 images. This dataset is split into three parts: a test set comprising 4518 images, a protection set with 25,000 images, and a reference set containing the remainder. The images are 112×112 pixels.

**Pets and Cars.**  Pets [43] contains 37 categories of animals, in which the set to be protected includes 3680 images and the test set contains 3669 images. Cars [28] contains 197 categories of automobiles, in which the set to be protected includes 8144 images, and the test set contains 8041 images. Similar to ImageNet-Subset, images are processed to be 224×224.

Table 5 summarizes the information about the datasets used for classification tasks. We delay the details of data preparation for the style mimicry task to Appendix B.5.

Table 5: Dataset details.

|  | Protection | Reference | Test | Categories | Balanced |
|---|---|---|---|---|---|
| CIFAR-10 | 40,000 | 10,000 | 10,000 | 10 | ✓ |
| CIFAR-100 | 40,000 | 10,000 | 10,000 | 100 | ✓ |
| ImageNet-Subset | 25,000 | 10,000 | 50,000 | 100 | ✓ |
| WebFace-Subset | 25,000 | 15,179 | 4,518 | 100 | ✗ |
| Cars | 8144 | - | 8041 | 197 | ✓ |
| Pets | 3680 | - | 3669 | 37 | ✓ |

## B.2 Protection

For CIFAR-10/100, ImageNet-Subset, and WebFace-Subset, we generate the availability attacks on the combination of the protection set and reference set to simulate the exact protection mechanism. The additional paired data are collected from the original and protected reference datasets. In Appendix C.11, we will investigate more protections whose generation does not involve a reference dataset and present additional results showing the consistent effectiveness of BridgePure against them.

For Cars and Pets, the protection generation of UC(-CLIP) is determined by the clustering of the protection dataset. The generated protection can be easily applied to unseen data. Thus, we collect additional paired data from the protected test dataset.

For style mimicry protection, we will detail the implementation of Glaze and Mist in Appendix B.5.

## B.3 BridgePure

**Training.** We train BridgePure from scratch using each paired dataset for 100,000 steps. The batch size is 256 for CIFAR-10, CIFAR-100; 32 for WebFace-Subset; 16 for ImageNet-Subset, Cars, Pets, and @*nulevoy*'s artwork. Training on CIFAR-10/100 and WebFace-Subset can run on a single NVIDIA L40S/RTX 6000 Ada GPU with 40 GB of memory. Training on ImageNet-Subset, Cars, and Pets can run on a single NVIDIA A100 GPU with 80 GB of memory. Training on artwork can run on 4 NVIDIA A100 GPUs in parallel. By default, we use the VE mode for bridge models and will compare VE and VP modes in Appendix C.10.

**Sampling.** We adopt a 40-step sampling for all evaluated datasets. As recommended by DDBM [76], the guidance hyper-parameter is set to 1 for VP bridge models and to 0.5 for VE bridge models.

## B.4 Evaluation for Classification

To evaluate the restoration of availability, we train classifiers on the original/protected/purified datasets (i.e., protection set in Table 5) and calculate their accuracy on the test set. If not otherwise stated, we train a ResNet-18 classifier for 120 epochs using an SGD optimizer with an initial learning rate of 0.1, a momentum of 0.9, and a weight decay of 0.0005. The learning rate decays by 0.1 at the 80th and 100th epochs. The batch size is 128. For ViT and CaiT, we use Adam optimizer with an initial learning rate of 0.0005. We follow the evaluation setting from Zhang et al. [74] for UC and UC-CLIP.

For contrastive learning, we train an encoder with the ResNet-18 backbone using SimCLR with a temperature of 0.5. The batch size is 512. We use an SGD optimizer with an initial learning rate of 0.5, a momentum of 0.9, and a weight decay of 0.0001. The learning rate scheduler is cosine annealing with a 10-epoch warm-up. The linear probing stage uses an SGD optimizer for 100 epochs with an initial learning rate of 1.0 and a scheduler that decays by 0.2 at 60, 75, and 90-th epochs.

## B.5 Style Mimicry

**Artwork.** After obtaining the artist's permission via email, we collect @*nulevoy*'s artwork from his homepage on *ArtStation*. The paintings are 1920×1080 pixels. Since Hönig et al. [21] verified that Stable Diffusion v2.1 without fine-tuning fails to generate paintings of @*nulevoy*'s style, it is reasonable to use these artworks for the style mimicry task.

**Protections.** Glaze v2.1 takes an image of any shape as input and outputs a modified image of the same shape. Since it is a closed-source tool that only supports Windows and macOS platforms, we process the paintings on a MacBook Pro with an M3 Max chip. The protected paintings have the same shape as the original ones. The protection intensity is *High* and the render quality is *Slowest*.

Mist takes square images and outputs images of the same shape. However, the max size it supports is 768×768. To preserve the object ratios in the painting and the image quality, we first resize the short edge of images to 768, center-crop them to square ones, and then feed them into Mist. The resulting protected paintings are 768×768 pixels.

**Mimicry pipeline.** We adopt the style mimicry implementation from Hönig et al. [21], which involves fine-tuning Stable Diffusion v2.1 [50] using a set of captioned paintings. For fine-tuning, the images are first center-cropped to 512×512 and their captions are auto-generated by a BLIP-2 model [31]. The fine-tuned model generates 768×768-pixel images based on predefined test prompts.

We randomly select 20 paintings from artist @*nulevoy* for fine-tuning and use the same 10 prompts[4] from Hönig et al. [21] to evaluate the mimicry performance. For Mist, the mimicry process performs center-cropping on the 768×768 squared images, while for Glaze, the mimicry process performs center-cropping on the original images.

**BridgePure implementation.** Assume a protection leakage consists of 5 or 10 pairs of original and protected artwork. To augment this dataset, we randomly crop the artwork to 512×512 pixels, generating a paired dataset with 1,000 pairs of paintings. BridgePure is then trained using this augmented paired dataset.

For the style mimicry task, the protected fine-tuning set comprises 20 paintings, which are center-cropped to 512×512 pixels from the protected outputs of Glaze or Mist. BridgePure sanitizes these images, and the purified outputs are subsequently fed into the mimicry pipeline.

## C  Additional Experiment Results

### C.1  Time Consumption

On our machine with NVIDIA A100 GPUs, training a BridgePure on CIFAR-10/100 costs around 22.5 hours with a single GPU, and that on ImageNet-Subset costs around 24 hours with a single GPU. For sampling a batch of 64 images from ImageNet-Subset with a single GPU, BridgePure costs 138 seconds on average while DiffPure ($t^*$=0.1) costs 165 seconds.

On one hand, we empirically observed that early-stopping could reduce the time cost in BridgePure training. For example, BridgePure-4K trained with 40K steps on WebFace-Subset, which only costs 450 minutes in training, recovers the test accuracy of EM/LSP/TAP-protected dataset to 87.88/87.87/87.61%. On the other hand, BridgePure follows an offline training scheme similar to other models—once trained, the model can purify an unlimited number of protected samples within the same domain. The additional computational overhead for each new sample is limited to inference cost only, which is minimal compared to the initial training. In other words, the purification cost for each image is amortized. Therefore, the training consumption shows no obstacle for malicious adversaries.

### C.2  Visualization of Sanitized Images

We show original, protected, and BridgePure-purified images from CIFAR-10 and WebFace-Subset in Figure 9. Although availability attacks make perturbations less noticeable by imposing norm constraints, upon zooming in and comparing the protected image with the original, one can observe slight differences. However, images purified by BridgePure are indistinguishable from the original to human eyes.

### C.3  Additional Generated Images in Style Mimicry Task

Figure 10 provides additional generated images in the style mimicry task investigated by Section 5.3. As discussed in Section 5.3, BridgePure eliminates the protective cloaks in the mimicry outputs.

---

[4]The prompts for style mimicry include "a mountain by nulevoy", "a piano by nulevoy", "a shoe by nulevoy", "a candle by nulevoy", "a astronaut riding a horse by nulevoy", "a shoe with a plant growing inside by nulevoy", "a feathered car by nulevoy", "a golden apple by nulevoy", "a castle in the jungle by nulevoy", and "a village in a thunderstorm by nulevoy".

## C.4 Purification Quality for Style Mimicry

Figure 11 compares the purification effects with recent methods, including GrIDPure [75], PDM [68], and NoisyUpscaling [21]. For both Glaze and Mist, BridgePure-10 effectively removes the protective cloaks, whereas other methods leave behind visually perceptible patterns.

Since PDM performs comparably to BridgePure, Figure 12 provides a comparison of the fine details in the purified paintings. PDM automatically smooths out sharp brushstrokes, whereas BridgePure preserves them perfectly. The preservation of these details is crucial for faithfully mimicking the artist's style. Our results demonstrate that BridgePure achieves superior purification performance, particularly in preserving fine details while effectively removing protection cloaks.

## C.5 Minor Protection Leakage

In previous tables, we report the results of BridgePure trained with protection leakage ranging from 500 to 4000 pairs. Figure 13 investigates the performance of BridgePure with less leakage, i.e., from 20 to 500 pairs, on CIFAR-10 and CIFAR-100 protected by LSP. For CIFAR-10, 100 pairs are sufficient for BridgePure to improve the test accuracy to 93%, while for CIFAR-100, BridgePure-100 only restores the accuracy to 50%, and BridgePure-200 improves it to 69%. This difference in purification performance with minor protection leakage is because CIFAR-100 has 10 times more categories, and thus, the leakage in each class is much less than that for CIFAR-10.

## C.6 Comparison with Augmentation-Based Methods and Protection Dilution

Figure 14 shows the detailed performance comparison with augmentation-based methods and protection dilution on CIFAR-10 and CIFAR-100, complementary to Figure 2. The augmentation-based methods include Cutout, Cutmix, Mixup, Gaussian Blur, Grayscale, JPEG Compression, bit depth reduction(BDR), and UEraser [45]. Regarding protection dilution, Dilution-4K means adding 4,000 unprotected images to the protected dataset and training a classifier using the combined data. On both CIFAR-10 and CIFAR-100, our BridgePure-0.5K (sky blue dots in figures) consistently surpasses these other methods (dots with other colors).

Furthermore, it is well known that availability attacks are sensitive to the dilution of clean images. That is, mixing some unprotected images into the protected dataset could improve the test accuracy of trained classifiers. However, protection leakage poses a much more severe risk than protection dilution since it exposes the protection mechanism. Table 6 compares BridgePure with dilution on CIFAR-10 and CIFAR-100. With the same number of accessible unprotected images, BridgePure shows much better availability restoration than dilution.

Table 6: Comparison between BridgePure and protection dilution. For example, Dilution-4K means adding 4,000 unprotected images to the protected dataset and training a classifier using the combined data.

|  |  | AR | DC | EM | GUE | LSP | NTGA | OPS | REM | TAP | Average |
|---|---|---|---|---|---|---|---|---|---|---|---|
| CIFAR10 | Dilution-0.5K | 36.6 | 46.4 | 43.6 | 45.8 | 48.5 | 54.7 | 54.0 | 42.5 | 71.3 | 49.3 |
|  | BridgePure-0.5K | 93.9 | 93.8 | 93.6 | 93.7 | 93.8 | 94.1 | 93.3 | 84.3 | 86.8 | 91.9 |
|  | Dilution-4K | 79.6 | 80.3 | 77.9 | 79.4 | 80.1 | 80.7 | 80.6 | 79.2 | 84.9 | 80.3 |
|  | BridgePure-4K | 93.6 | 93.8 | 93.9 | 93.8 | 93.9 | 93.9 | 93.5 | 93.5 | 92.9 | 93.7 |
| CIFAR100 | Dilution-0.5K | 15.8 | 51.9 | 14.0 | 31.3 | 13.5 | 15.9 | 28.2 | 17.7 | 27.7 | 24.0 |
|  | BridgePure-0.5K | 67.5 | 73.7 | 73.2 | 72.7 | 73.3 | 69.1 | 74.2 | 66.5 | 62.8 | 70.3 |
|  | Dilution-4K | 48.7 | 64.4 | 40.6 | 52.8 | 45.2 | 46.3 | 52.1 | 42.0 | 54.6 | 49.6 |
|  | BridgePure-4K | 72.4 | 74.0 | 73.5 | 73.9 | 74.6 | 74.2 | 74.2 | 73.0 | 71.0 | 73.4 |

## C.7 Evaluation with More Network Architectures

In Table 7, we evaluate the purified CIFAR-10 datasets for classification using various network architectures, including SENet-18 [22], MobileNet v2 [53], DenseNet-121 [23], ViT [11], and CaiT [63]. It shows that the purification effect of BridgePure is consistent across networks.

Table 7: We evaluate BridgePure-1K-sanitized CIFAR-10 datasets using different network architectures. The baseline is trained on unprotected data.

| | Baseline | AR | DC | EM | GUE | LSP | NTGA | OPS | REM | TAP |
|---|---|---|---|---|---|---|---|---|---|---|
| SENet-18 | $94.00_{\pm0.18}$ | $91.79_{\pm0.26}$ | $93.78_{\pm0.12}$ | $93.73_{\pm0.11}$ | $93.77_{\pm0.31}$ | $93.96_{\pm0.15}$ | $93.96_{\pm0.18}$ | $93.28_{\pm0.16}$ | $92.32_{\pm0.36}$ | $87.37_{\pm0.10}$ |
| MobileNet v2 | $90.60_{\pm0.29}$ | $87.63_{\pm0.44}$ | $90.29_{\pm0.11}$ | $90.17_{\pm0.18}$ | $90.40_{\pm0.10}$ | $90.43_{\pm0.15}$ | $90.73_{\pm0.42}$ | $90.32_{\pm0.12}$ | $89.03_{\pm0.38}$ | $84.54_{\pm0.24}$ |
| DenseNet-121 | $94.44_{\pm0.15}$ | $92.24_{\pm0.16}$ | $94.32_{\pm0.23}$ | $93.92_{\pm0.29}$ | $94.11_{\pm0.14}$ | $94.07_{\pm0.16}$ | $94.37_{\pm0.10}$ | $93.74_{\pm0.12}$ | $92.93_{\pm0.24}$ | $87.75_{\pm0.26}$ |
| ViT | $84.80_{\pm0.15}$ | $84.61_{\pm0.27}$ | $84.48_{\pm0.50}$ | $84.26_{\pm0.11}$ | $83.94_{\pm0.52}$ | $84.80_{\pm0.39}$ | $84.82_{\pm0.21}$ | $84.89_{\pm0.43}$ | $83.95_{\pm0.08}$ | $80.05_{\pm0.34}$ |
| CaiT | $82.73_{\pm0.23}$ | $82.53_{\pm0.18}$ | $82.20_{\pm0.78}$ | $81.91_{\pm0.45}$ | $81.58_{\pm0.43}$ | $82.55_{\pm0.21}$ | $82.71_{\pm0.15}$ | $82.41_{\pm0.25}$ | $81.90_{\pm0.15}$ | $78.09_{\pm0.33}$ |

Table 8: Transferablity of BridgePure-4K across CIFAR-10 and CIFAR-100. For example, CIFAR-100 $\rightarrow$ CIFAR-10 means BridgePure is trained using protection leakage of CIFAR-100 and is used to purify protected CIFAR-10. Here $s = 0.33$ and $\beta = 0$.

| Transfer | | AR | DC | EM | GUE | LSP | NTGA | OPS | REM | TAP |
|---|---|---|---|---|---|---|---|---|---|---|
| CIFAR-100 $\rightarrow$ CIFAR-10 | Protected | $13.52_{\pm0.63}$ | $15.10_{\pm0.81}$ | $23.79_{\pm0.13}$ | $12.76_{\pm0.44}$ | $13.85_{\pm0.96}$ | $12.87_{\pm0.23}$ | $13.67_{\pm1.80}$ | $20.96_{\pm1.70}$ | $9.51_{\pm0.67}$ |
| ($94.01_{\pm0.15}$) | Purified | $32.16_{\pm0.36}$ | $37.33_{\pm3.05}$ | $63.90_{\pm0.80}$ | $27.65_{\pm0.73}$ | $90.26_{\pm0.26}$ | $65.94_{\pm1.02}$ | $93.43_{\pm0.27}$ | $30.22_{\pm0.78}$ | $78.18_{\pm0.55}$ |
| CIFAR-10 $\rightarrow$ CIFAR-100 | Protected | $2.02_{\pm0.12}$ | $36.10_{\pm0.67}$ | $6.73_{\pm0.12}$ | $19.50_{\pm0.48}$ | $2.56_{\pm0.16}$ | $1.51_{\pm0.22}$ | $12.18_{\pm0.52}$ | $7.07_{\pm0.19}$ | $3.59_{\pm0.12}$ |
| ($74.27_{\pm0.45}$) | Purified | $13.74_{\pm0.26}$ | $53.22_{\pm0.78}$ | $42.96_{\pm0.50}$ | $33.55_{\pm0.62}$ | $54.33_{\pm0.93}$ | $28.91_{\pm1.53}$ | $58.18_{\pm1.74}$ | $15.89_{\pm0.14}$ | $41.75_{\pm0.32}$ |

## C.8 Transferability across Protections

Although Table 4 demonstrates that randomly mixing multiple protection mechanisms fails to hinder an adversary from deriving an effective BridgePure, we consider a *different* scenario in which the adversary collects some additional data $\mathcal{D}_a$ but calls a different protection mechanism $\mathcal{P}'$, derives a BridgePure using such pairs, and then purifies a dataset protected by $\mathcal{P}$. In this case, the purification ability of BridgePure reflects its transferability across different protections.

On classification tasks, Figure 15 shows that BridgePure has limited transferability across protections, and advanced purification relies on the awareness of the underlying mechanism for the protected data.

On style mimicry tasks, Figure 16 shows that BridgePure trained on Mist effectively purifies Glaze-protected paintings, and BridgePure trained on Glaze largely reduces Mist-patterns in the generated paintings.

In summary, although BridgePure exhibits varying degrees of cross-protection transferability on different tasks, this does not undermine the main claim of this paper—namely, that the protection leakage outlined in the threat model poses a serious security risk.

## C.9 Transferability across Data Distributions

In our threat model, we assume the additional dataset $\mathcal{D}_a$ is sampled from the same distribution as that for $\mathcal{D}$. Now we consider a *different* scenario where an adversary cannot collect additional data from the same distribution but from another distribution, e.g., $\mathcal{D}$ is from CIFAR-10 and $\mathcal{D}_a$ is from CIFAR-100, or vice versa.

On classification tasks, we investigate the influence of such distribution mismatch on the purification performance of BridgePure in Table 8. When BridgePure is trained on pairs from CIFAR-100 and is used to purify protected images from CIFAR-10, the accuracy for OPS and LSP is over $90\%$, but that for other protections is lower than $80\%$. When BridgePure is trained on pairs from CIFAR-10 and is used to purify protected images from CIFAR-100, the accuracy for all nine protections is lower than $60\%$. The reasons why BidgePure transfers well from CIFAR-100 to CIFAR-10 for LSP and OPS could be (1) OPS and LSP create rather regular patterns for protection while other methods generate irregular patterns (see Figure 9); (2) CIFAR-100 is more fine-grained than CIFAR-10 and thus CIFAR-100 pairs might cover the protection mechanism for CIFAR-10.

On style mimicry tasks, we train BridgePure using painting pairs by the renowned Impressionist artist Claude Monet and use it to purify protected @*nulevoy*'s artwork. Figure 17 shows that the generated images are free of any protective patterns, indicating that BridgePure transfers well across different art styles.

## C.10 Comparison between VE and VP Bridges

DDBM [76] supports two modes for the diffusion process: variance exploding (VE) and variance preserving (VP). Figure 18 compares the performance of VE and VP bridges on CIFAR-10 and CIFAR-100. When facing REM and TAP attacks on CIFAR-100, the VE bridge consistently outperforms the VP bridge for two values of $s$. In other cases, the purification effects of the two modes are comparable. Therefore, we adopt the VE bridge as the default setting in this paper.

Table 9: Purification performance on CIFAR-10 and CIFAR-100 against EMC*, OPS* and TAP* protections.

|  | EMC* | OPS* | TAP* | EMC* | OPS* | TAP* |
|---|---|---|---|---|---|---|
|  | CIFAR-10 ($94.01_{\pm0.15}$) | | | CIFAR-100 ($74.27_{\pm0.45}$) | | |
| Protected | $13.05_{\pm0.54}$ | $12.01_{\pm0.97}$ | $7.68_{\pm0.50}$ | $1.41_{\pm0.11}$ | $12.44_{\pm0.66}$ | $3.24_{\pm0.32}$ |
| BridgePure-0.5K | $93.98_{\pm0.17}$ | $92.99_{\pm0.02}$ | $80.20_{\pm0.28}$ | $74.46_{\pm0.16}$ | $73.70_{\pm0.14}$ | $59.31_{\pm0.38}$ |
| BridgePure-1K | $94.06_{\pm0.10}$ | $93.52_{\pm0.30}$ | $82.44_{\pm0.40}$ | $74.54_{\pm0.17}$ | $74.26_{\pm0.16}$ | $63.79_{\pm0.29}$ |
| BridgePure-2K | $93.85_{\pm0.17}$ | $93.14_{\pm0.23}$ | $90.55_{\pm0.23}$ | $74.22_{\pm0.39}$ | $74.38_{\pm0.25}$ | $63.01_{\pm0.56}$ |
| BridgePure-4K | $93.95_{\pm0.15}$ | $93.92_{\pm0.08}$ | $93.07_{\pm0.19}$ | $74.00_{\pm0.39}$ | $74.36_{\pm0.38}$ | $69.92_{\pm0.13}$ |

## C.11 More Discussion on Protection for Additional Data

Note that our threat model assumes that the protection mechanism $\mathcal{P}$ can generate (unprotected, protected) pairs using only the additional data $\mathcal{D}_a$. While some availability attacks such as LSP, UC, UC-CLIP, Glaze, and Mist are precisely examined in this way, some other attacks may not fit exactly into the threat model. For example, EM and REM generate sample-wise protection on the dataset they optimize. Thus performing the protections on $\mathcal{D}$ and $\mathcal{D}_a$ separately may result in different protection mechanisms.

To ensure that the protection is consistent for $\mathcal{D}$ and $\mathcal{D}_a$, we generate the protection using both $\mathcal{D}$ and the reference set from which $\mathcal{D}_a$ is sampled and evaluate the attacks in Tables 1 and 2. This may pose a slightly stronger protection leakage that allows an adversary to directly obtain the additional pairs. Here we consider three additional variants of the attacks we considered previously and allow access to $\mathcal{D}_a$ only:

- EMC*: We generate class-wise EM protection [24] using the 40K images to be protected and apply the protection to additionally collected data.

- OPS*: Similar to EMC*, we generate OPS protection [67] using the 40K images to be protected and apply the protection to additionally collected data.

- TAP*: The reference classifier is trained on the 40K images, and the protection for additional data is to search adversarial examples for this classifier [14].

We evaluate these three protections on CIFAR-10 and CIFAR-100 in Table 9 and the results confirm the potent purification ability of BridgePure that is consistent with the previous results in Section 5.2.

# D Countermeasures and Broader Impacts

## D.1 Possible Countermeasures against Protection Leakage

**Service provider side.** To prevent malicious adversaries from training a powerful BridgePure model, the most effective strategy is to restrict their access to protection leakage (paired data) derived from unauthorized sources. While it may be impossible to stop adversaries from acquiring a limited number of unprotected images, the critical safeguard is to limit their ability to use the protection mechanism (e.g., an API) to generate the corresponding protected versions. To implement this, we propose the following recommendations:

- Include special parameters or random seeds in open-sourced methods to control reproducibility. In real-world deployment, such configurations should prevent malicious adversaries from fully replicating the protection algorithm.

- Avoid offline protection services, such as standalone applications, as they lose control over the invocation of the protection mechanism and cannot prevent protection leakage. Offline data protection services should not guarantee strong security.
- Incorporate identity and data ownership verification into protection services. For example, in the case of artistic style protection, users should be required to declare and prove copyright ownership of the artwork to be protected, subject to provider review. The service should maintain a registry of verified styles and enforce that: (1) No single style can be registered by multiple users. (2) No single user can register multiple, conflicting styles. (3) Each user may only protect artworks consistent with their registered style.

An alternative line of defense against the BridgePure threat is to design protection mechanisms that are resilient to its purification capabilities. However, to the best of our knowledge, no existing availability attack or copyright protection method has been proposed that can effectively resist purification techniques based on diffusion models. Given that BridgePure directly learns the transformation between distributions—rather than relying on the traditional noise-adding and denoising pipeline—we believe that developing robust protection methods specifically against BridgePure represents a more compelling and challenging direction for future research, with promising potential for broad real-world applications.

**Service user side.** For protection service users, the most straightforward defense is to minimize leaking unprotected images and choose trustworthy protection service providers. We also believe the following directions show promise as additional defense strategies:

- Preemptive defenses: Users can develop adversarial preprocessing techniques that make their images inherently resistant to BridgePure attacks. One approach could involve gradient-based methods that require differentiating through the stochastic purification process, presenting an interesting direction for future research.
- Strategic decoys: Users can strategically release crafted decoy images designed to poison the attacker's training data, reducing BridgePure's effectiveness on their actual protected content. Potential approaches include gradient-based perturbation search, visible or invisible watermarking, and inducing significant domain shifts, among others.

The effectiveness of these mitigation strategies requires further investigation, and we look forward to investigating further in future work.

## D.2   Broader Impacts

This research focuses on the reliability of data protection methods in real-world scenarios. Through the deployment of BridgePure, we discovered that limited protection leakage can lead to the failure of existing protection mechanisms. Our findings have profound implications for the community. It underscores the urgent need for more resilient data protection frameworks. Additionally, it informs researchers and practitioners about the risks associated with current black-box protection approaches, fostering the development of more secure methodologies. Finally, it empowers data owners and service users by increasing awareness of the potential weaknesses in protection systems, helping them make more informed decisions when sharing sensitive data.

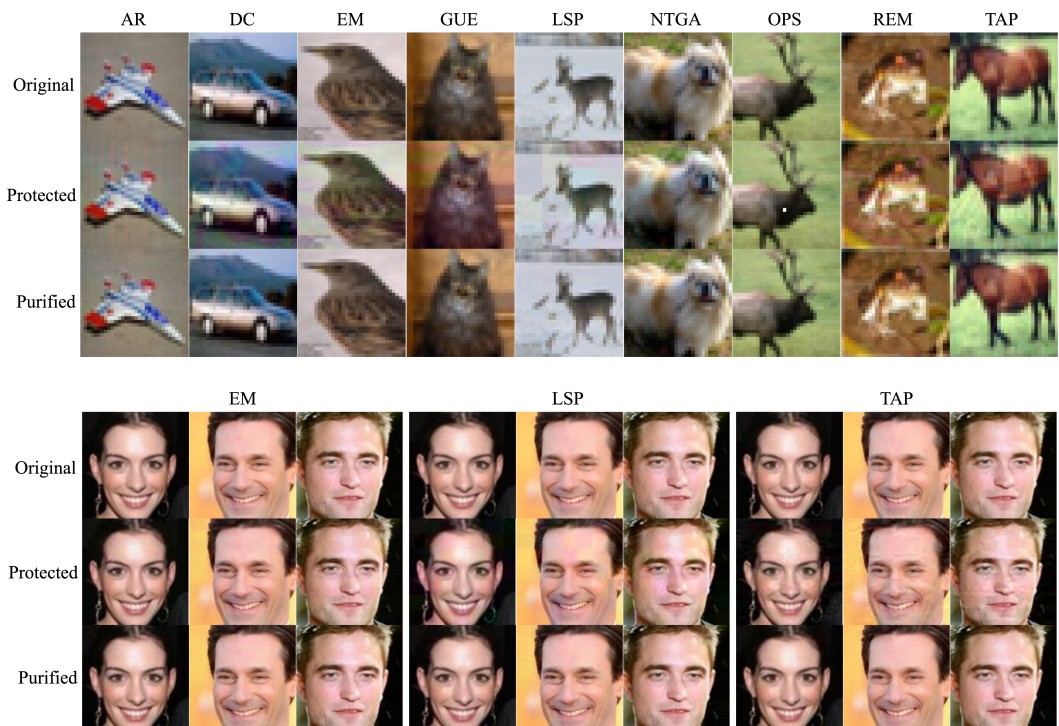

Figure 9: Visualization of our BridgePure-1K on CIFAR-10 (**top**) and WebFace-Subset (**bottom**).

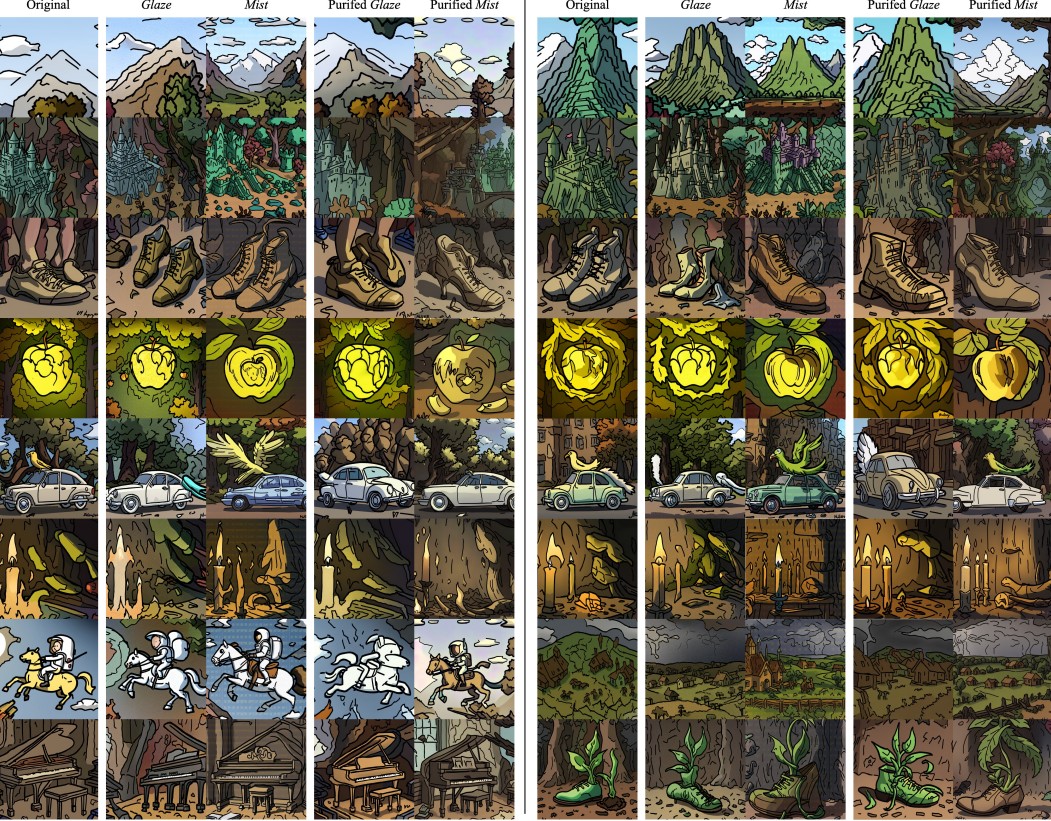

Figure 10: Additional results to Figure 7. Performance of BridgePure-5 (**left**) and BridgePure-10 (**right**) for style mimicry.

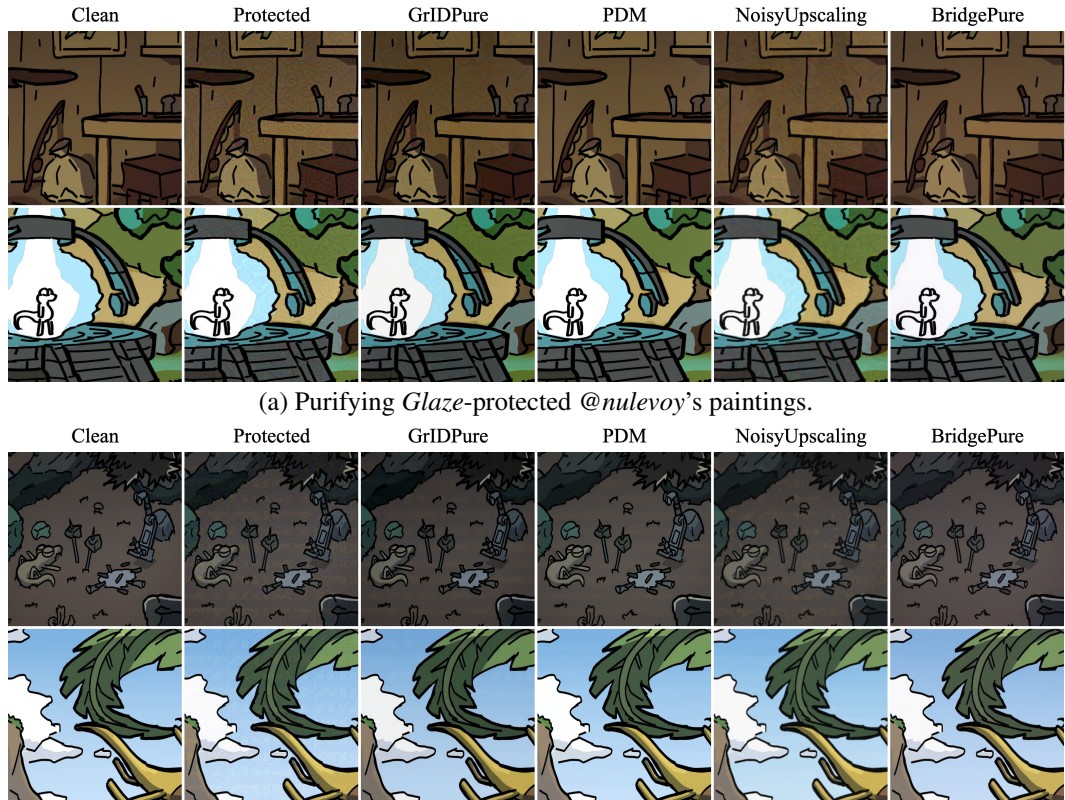

(a) Purifying *Glaze*-protected @*nulevoy*'s paintings.

(b) Purifying *Mist*-protected @*nulevoy*'s paintings.

Figure 11: Paintings purified by recent purification methods and BridgePure-10.

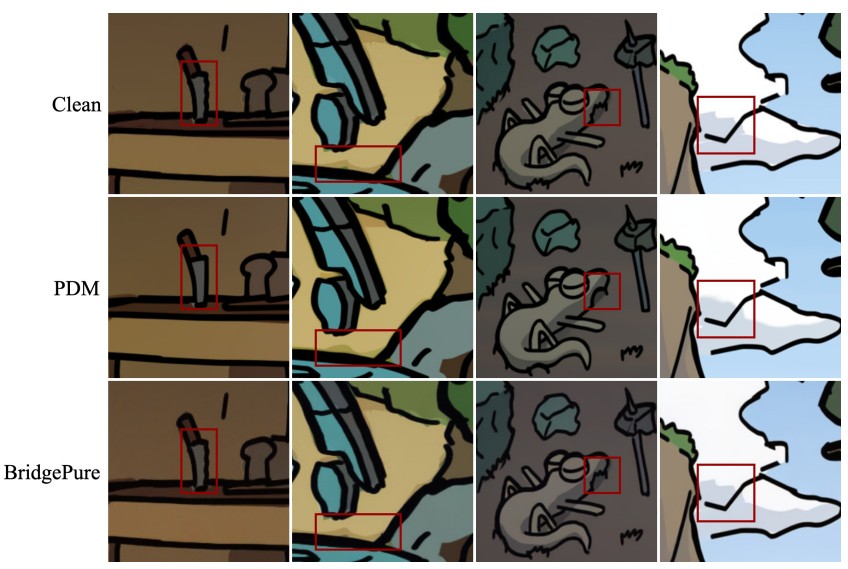

Figure 12: Comparison of purified painting details (cropped from Figure 11) between PDM and BridgePure. Red boxes emphasize the details that PDM blurs.

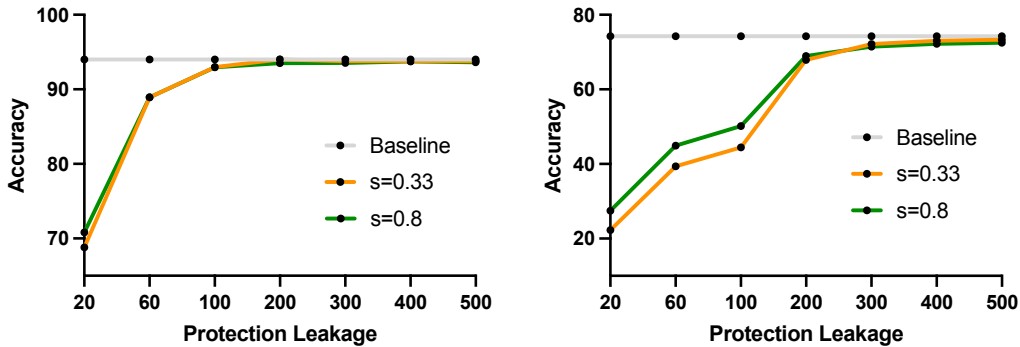

Figure 13: Purification performance of BridgePure with small protection leakages to purify LSP-protected CIFAR-10 (**left**) and CIFAR-100 (**right**). Here $\beta = 0$ and $s \in \{0.33, 0.8\}$.

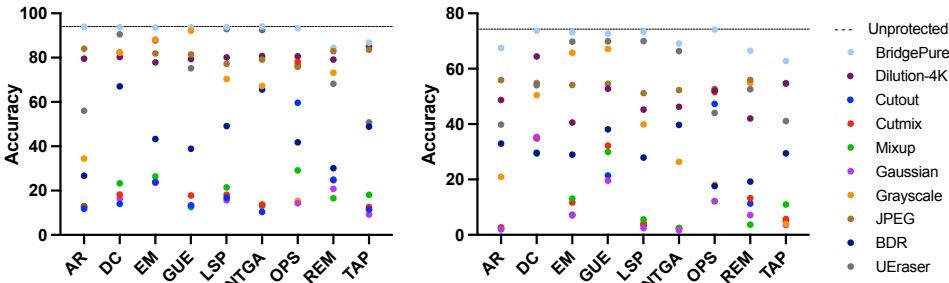

Figure 14: Performance comparison with augmentation-based methods, and protection dilution on CIFAR-10 (**left**) and CIFAR-100 (**right**). The sky blue dots show the performance of BridgePure-0.5. Dots with other colors stand for other circumvent methods. The dashed lines represent the unprotected baselines. The higher the dots, the better the accuracy recovery.

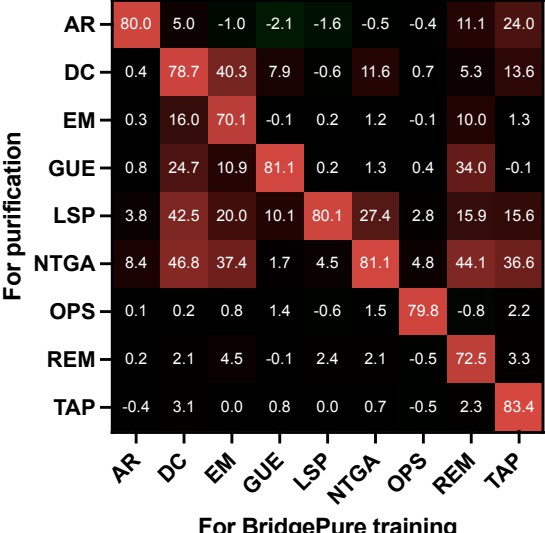

Figure 15: Transferablity of BridgePure-4K across different protections on CIFAR-10. The **x-axis** represents the protection leakage on which BridgePure is trained. The **y-axis** represents the protected dataset to which the pre-trained BridgePure is applied for purification. Each cell shows an improvement in test accuracy compared to the unpurified dataset. Here $s = 0.33$ and $\beta = 0$.

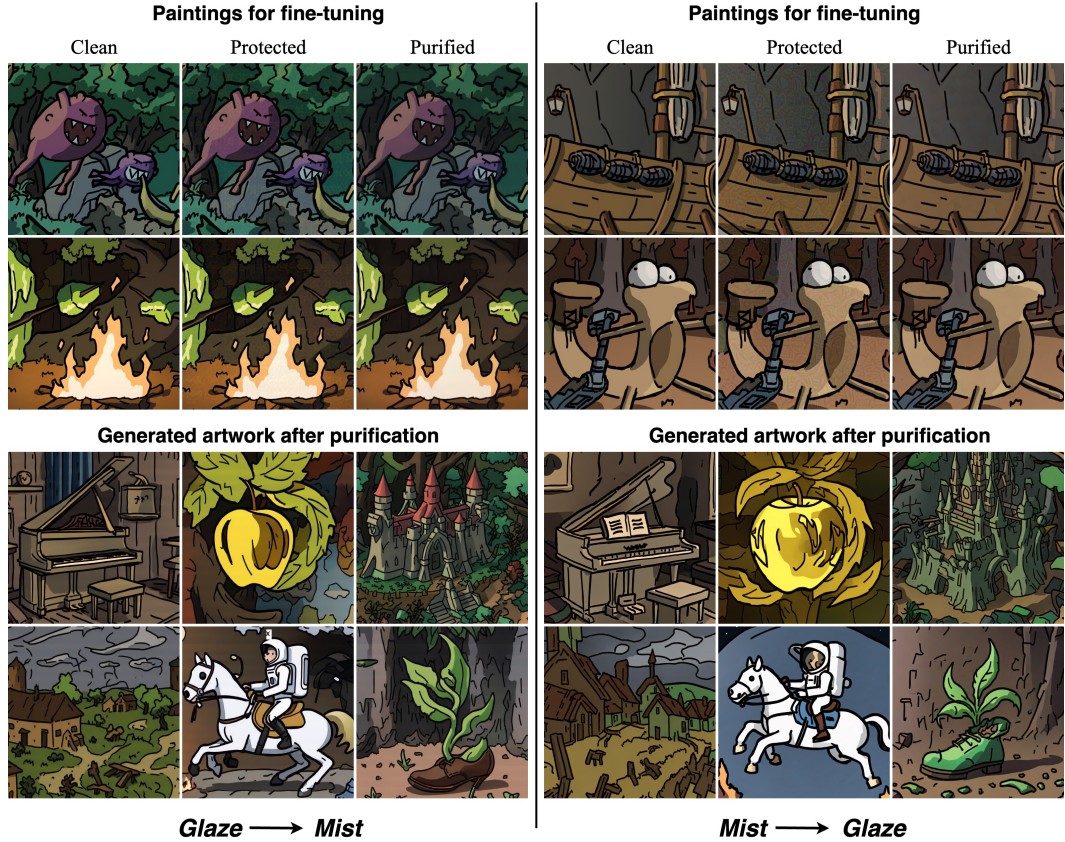

Figure 16: Transferability across style mimicry protections. **Left:** Using BridgePure-10 trained with (clean, Glaze-protected) pairs to purify Mist-protected paintings. **Right:** Using BridgePure-10 trained with (clean, Mist-protected) pairs to purify Glaze-protected paintings. **Top:** Clean paintings by @*nulevoy*, protected ones, and BridgePure-purified ones. **Bottom:** Mimicked artwork by prompting the Stable Diffusion v2.1 that is fine-tuned on the BridgePure-purified paintings.

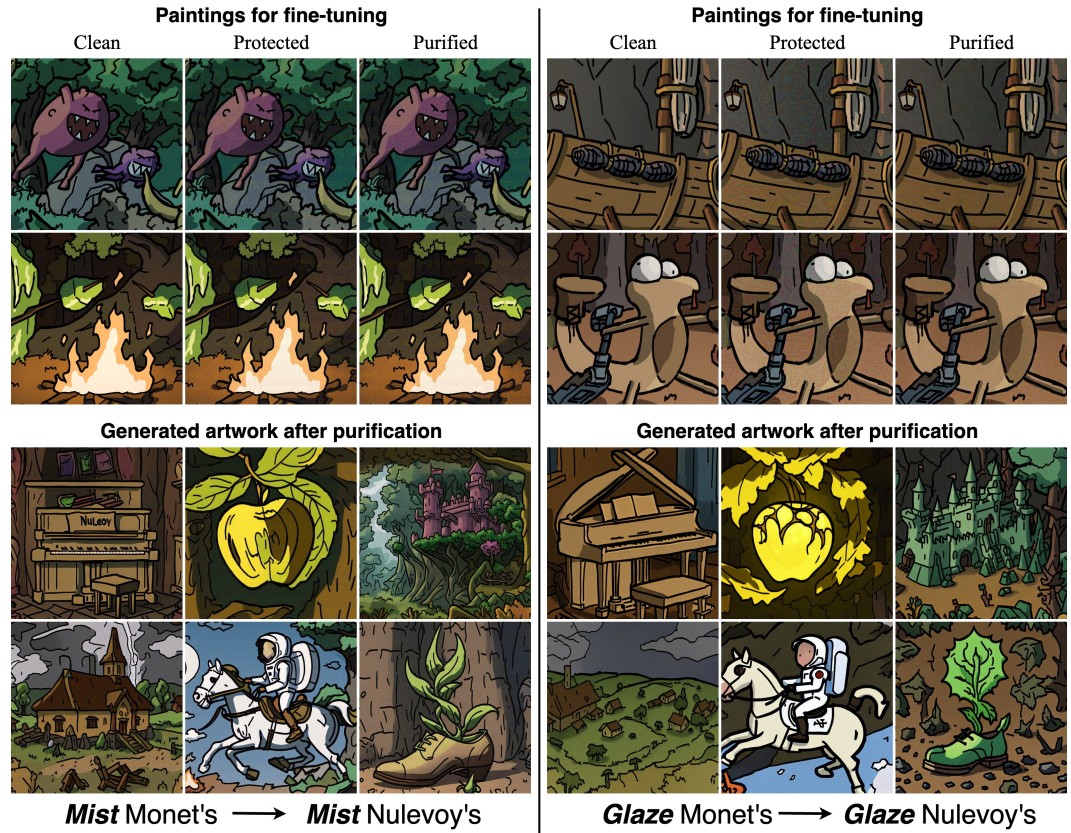

Figure 17: Transferability across datasets for style mimicry. We train BridgePure-10 on Monet's paintings and use it to purify *@nulevoy*'s protected artwork. **Left:** Both Monet's and *@nulevoy*'s paintings are protected by Mist. **Right:** Both Monet's and *@nulevoy*'s paintings are protected by Glaze. **Top:** Clean paintings by *@nulevoy*, protected ones, and BridgePure-purified ones. **Bottom:** Mimicked artwork by prompting the Stable Diffusion v2.1 that is fine-tuned on the BridgePure-purified paintings.

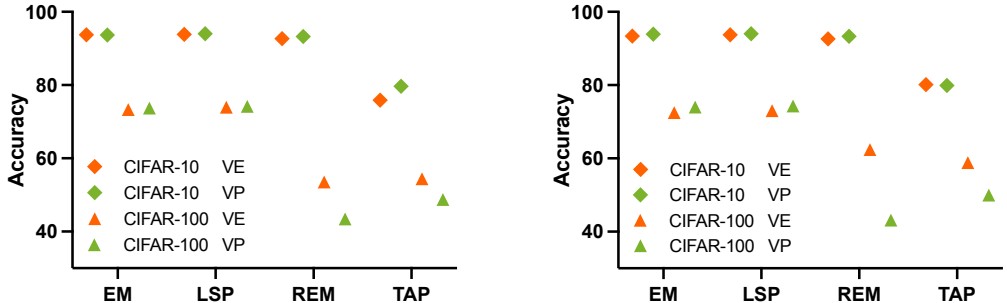

Figure 18: Comparison between VP and VE modes of BridgePure-1K with $s = 0.33$ (**left**) and with $s = 0.8$ (**right**). Here $\beta = 0$.

