# OpenReview forum: "BridgePure: Limited Protection Leakage Can Break Black-Box Data Protection"
_NeurIPS.cc/2025/Workshop/Reliable_ML — NeurIPS 2025 - Reliable ML Workshop_

### Official Review · Reviewer_mY6c · 2025-09-14
**Review of NeurIPS 2025 Workshop Reliable ML Submission36 -  BridgePure: Limited Protection Leakage Can Break Black-Box Data Protection**

**Rating:** 8
**Confidence:** 2

**Review:**

# Summary

The paper introduces a novel and practical threat model against black-box data protection services. These services, modeled as APIs for generating "unlearnable examples," allow users to modify their data to prevent unauthorized machine learning model training. The authors claim that if an adversary can obtain a small number of (unprotected, protected) data pairs (e.g., by using the public protection service on some publicly available, unprotected data from the same distribution), they can effectively break the protection for all other data from that distribution.

To demonstrate this vulnerability, the paper proposes “BridgePure”, a method that trains a Denoising Diffusion Bridge Model (DDBM) on these leaked pairs. This model learns the inverse mapping from the protected data distribution back to the original, unprotected distribution. The main results show that BridgePure can restore the utility ("availability") of protected datasets for both classification and style mimicry tasks. Across numerous datasets (CIFAR-10/100, ImageNet-Subset) and a wide array of availability attacks, BridgePure consistently outperforms existing purification baselines, often closing the availability gap entirely with as few as 500 to 4000 leaked pairs.

# Strengths
- I think the main contribution is a novel and very realistic threat model against black-box data protection APIs, where an adversary may have access to a small, unprotected in-distribution data, and black-box access to the data protection API. This is an important conceptual contribution overlooked by prior works
- The empirical validation is reasonably thorough, comparing against many popular data protection algorithms and on standard benchmark datasets. The results demonstrate a consistent improvement over baselines.
- The paper is well-written, clearly structured, and easy to follow.

# Weaknesses / Limitations
1. The proposed model relies on two key assumptions:
    - The data protection algorithm is static, e.g., the adversary has black-box access to the _exact_ same algorithm used to protect the data.
    - The adversary has access to a small number of unprotected, in-distribution datapoints.

While the first one may be reasonable, it seems unrealistic to assume the attacker has access to the exact private dataset.

2. The utility of BridgePure seems to depend on the protection-dependent hyperparameters. The presented results are tuned at different hyperparameters for different protection algorithms, which may not be realistic for real-world applications, where we cannot perform hyperparameter tuning against unprotected data.

# Suggestions for Authors
1. The empirical results would be even more convincing if the small, unprotected data was slightly out-of-distribution, e.g., we want to purify CIFAR-100 data given a small number of CIFAR-10 or ImageNet images.
1. I am unfamiliar with the popular protection attacks, but how much do the purification results depend on having the exact protection algorithm? I can imagine a scenario where some hyperparameters or the random seed of the black-box protection APIs are changed from day-to-day, so the exact same algorithm may not be available. How effective is BridgePure in this scenario? In other words, could this be a way to mitigate the threat model?
1. It would be interesting to compare the performance across different protection algorithms for fixed sets of hyperparameters (use the same parameters for different protection algorithms), since we cannot do hyperparameter tuning on a large, clean dataset in practical situations.

---

### Official Review · Reviewer_sWEL · 2025-09-20
**BridgePure: Limited Protection Leakage Can Break Black-Box Data Protection**

**Rating:** 8
**Confidence:** 3

**Review:**

Summary: In this paper, the authors investigate a practical threat to black-box data-protection methods. They introduce BridgePure, a diffusion-based denoising bridge conditioned on the protected image that using a small set of leaked (original, protected) pairs learns an approximate inverse of the protection. Across several datasets and protection/attack families, BridgePure consistently outperforms prior purification approaches.

Strengths:The paper asks a timely, practical question—can a small leak of paired data undermine black-box protection tools? It backs this up with solid empirical coverage across multiple datasets and protection/attack types, showing consistent gains. The core idea—a diffusion “bridge” conditioned on the protected image—is straightforward, clear, and likely easy for others to adapt.

Weaknesses: I’m unsure how an attacker would actually get a small, in-distribution set of (original, protected) pairs.

Suggestion: Run a stress test where the leaked pairs are slightly out-of-distribution (different camera/artist/time) and add a quick note on how an attacker might realistically get those pairs (e.g., rate limits, costs, ToS).